# Processing supramolecular framework for free interconvertible liquid separation

Guohua Zhang [1], Bingyu Li [1], Yan Zhou [1], Xiaofei Chen [1], Bao Li [1], Zhong-Yuan Lu [1]* & Lixin Wu [1]*

Nanoporous structures constructed by small molecular components exhibited vigorous materials potentials. While maintianing uniform porosity and functional properties, more applicable processing methods for the solid powders need to be considered and the improvement of binding interactions represents a preferable approach for structural flexibility. Here, by combining ionic interaction and host-guest inclusion, we constructed flexible supramolecular frameworks composing of inorganic polyanionic clusters, cationic organic hosts, and a bridging guest. The formed layer framework structure assemblies grew into nano-fibers and then supramolecular gels, donating highly convenient processability to porous materials. A simple spin-coating generated a new type of liquid separation membranes which showed structural stability for many liquids. The surface properties can be facilely modulated via filling a joystick liquid and then a hydrophilic/hydrophobic liquid into the porous frameworks, providing in-situ consecutive switchings for cutting liquids. This strategy extends the potential of flexible supramolecular frameworks for responsive materials in the laboratory and in industry.

[1] State Key Laboratory of Supramolecular Structure and Materials, College of Chemistry, Jilin University, Changchun 130012, P. R. China. *email: luzy@jlu.edu. cn; wulx@jlu.edu.cn

**D**uring the construction of ordered porous structure materials at molecular scale, coordination and covalent bonds that dominate binding affinity including bond number, bond angle and strength, as well as structural symmetry, play a key role in diverse fantastic frameworks with uniform porosity comprising of molecular components like metal organic frameworks (MOFs) and covalent organic frameworks (COFs)[1,2]. While bringing about irreplaceable advantages in holding structure's beauty, pore's rigidity and stability as well as geometrical diversity[3–7], the fixed connections also lead to crystalline solid and powder, which largely restrict their processability. For example, macroscopic/mesoscopic scale structuring of porous materials based on coordination and covalent bond from 0D to 3D structures depends on the deposition/growth of fine crystals with the assistance of template or support[8]. Strenuous efforts have been made to achieve soft behavior of porous crystalline solids[9–11] by the regulation of external physical and chemical environments such as pressure[12], thermal[13], polarity[14], and adsorption[15], yet the yielded phase performance, for instance breathing and swelling, only occurs in a small range discontinuously. In contrast to this, the intermolecular interaction that does not require strict angle restriction could be one of strong competitive forces as it can provide enough structural flexibility for various scale assembly and processing[16–20]. However, the interfacial energy that makes sub-interaction unavoidable and often drives molecules to get into a close packing state has to be considered[21]. Thus, during searching soft framework structures at macroscopic/mesoscopic scale, both binding forces used in rigid frameworks and the molecular units used for self-assemblies in solution have to be improved appropriately.

Convenient separations of oily components from effluents and quickly taking off water remnants from oil-products not only involve in chemical processing, but also become a pivotal integrant in dealing with serious environmental pollutions and recycling of valuable resources[22,23]. Depending on the dispersion state of the components, various physical and chemical methods are employed. Although there are several known techniques showing merits, higher efficient and more convenient approaches are still highly desired. Following the filtration and molecular sieving techniques, quick membrane separation of nonwetting fluids has become one of promising strategies with easy operation and energy-saving and emission reduction properties. It has been proved that the porous materials with solvent trapping ability represent one of the alternative approaches to design oil/water

separation membranes with smart switching and antifouling properties[24–28]. In this regard, the important nature of low molecular weight supramolecular gels[29] bearing various micro-/nano-scaled self-assembly structure can, in principle, be used to improve surface hydrophilicity/hydrophobicity of an oil-water separation membrane by simply spreading the gel assemblies onto solid surface of devices as well. Up to date, the supramolecular gel assemblies as independent materials have not yet been used to fabricate an independent separation membrane or to cover on the whole surface of a supporting substrate because of their unsettled structural stability. Actually, while renouncing the unfavorable aspects of supramolecular assemblies, their applicable surface qualities that the elaborate structure versatility brings about are discarded at the same time. Moreover, if only the structural strength can be improved greatly while the surface property becomes adaptable to the environment, it is possible to fabricate unusual materials that integrate the superiority of self-assembled structure and intermolecular interaction. Therefore, a rational design by incorporating framework assembly into supramolecular gels will help to realize the processability of porous materials for liquid separation.

In this context, we herein present a self-assembly strategy on the construction of supramolecular frameworks which are comprised of hydrophilic inorganic cluster that is pre-combined ionically with hydrophobic organic ligands for bond-control and the hydrophobic linker (Fig. 1a). The formed crosslinked organic-inorganic framework structure driven by electrostatic and host–guest interactions not only affords stable fibrous self-assemblies with flexible porosity in gel state, but also alters the surface property of the prepared gel membrane quickly through the wetting of intermediate fluids to the amphiphilic channels in framework structure (Fig. 1b). Thus, water and oil as well as other immiscible liquids can be nonstop switched to flow past the dexterous membrane for interconvertible separations. The unrestricted change over for selective passing through of organic fluids and aqueous solutions is much suitable for consecutive separation systems, and thus, wide potentials can be expectable because the one-step formed membranes will no longer rely on the materials and shapes of supports when the supramolecular framework gels are applied.

## Results

### Synthesis and structural characterization of ionic complex hosts.
Hydrophilic polyoxometalates (POMs), as a type of

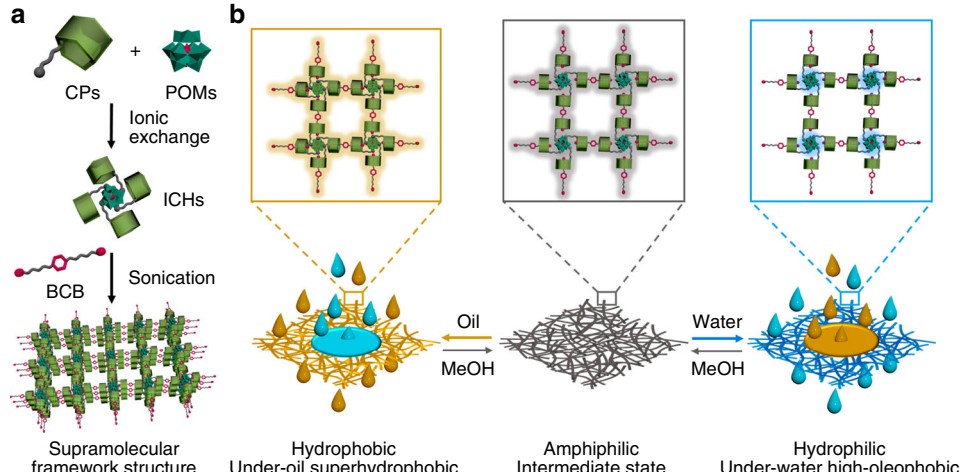

**Fig. 1 Supramolecular framework assembly in gel membrane for switchable liquid separation. a** Schematic formation of supramolecular framework driven by pre-ionic and post host–guest interaction, where CPs denotes cationic pillar[5]arenes, ICHs means the ionic complex hosts. **b** Modulation of membranes via intermediate state (gray) for changing over in oil (yellow) and water (blue) separation.

metal-oxide molecular clusters with uniform nano-size and topologic architecture, are ideal inorganic nodes for the design of the target flexible framework assemblies[30,31]. The delocalized negative charges enable POMs to link with organic counter-cations via ionic interaction, and the formed electrostatic complexes with weakened polarity become soluble in weak polar organic phase due to the covering hydrophobic organic components shielding the hydrophilicity of POMs[32]. The ionic complexes are prepared following a similar method previously reported[33], for instance, when sphere like polyanionic clusters such as $[SiW_{12}O_{40}]^{4-}$ (abbreviated as SiW) are mixed with cationic pillar[5]arenes, like (4-triethylammonium)butoxylmethoxyl pillar[5]arene bromide (abbreviated as TBP·Br) at a molar ratio of 1:4 in methanol, a full precipitation occurs after hours of stirring at room temperature (25 °C). Further washing the precipitate with water and methanol thoroughly gives the purified product (TBP)$_4$SiW. Systematic characterizations of $^1$H NMR, $^{13}$C NMR, and MALDI-TOF MS spectra as well as elemental analysis demonstrate the charge-compensated ionic complex. The chemical shifts ascribing to protons from the ammonium head of TBP in the formed complexes show obvious moving and broadening in comparison to the group at its isolated state, indicating the tight ionic interaction[34]. Considering the bands moving of the absorptions ascribing to W−O−W and W=O vibrations of SiW before and after the complexation in IR spectra, strong electrostatic interaction between SiW and TBP is further confirmed[35]. The results from mass spectrum and elemental analysis are in good agreement with the proposed molar ratio between organic and inorganic components. X-ray photoelectronic spectrum (XPS) demonstrates the unchanged structure of SiW after the ionic complexation as well. Similarly, the preparation of series complexes comprising of different clusters and cationic pillar[5]arenes is performed following the same route and the chemical formulae are identified. All structural characterization data are summarized in Electronic Supplementary Information (Supplementary Figs. 1−34).

**Host-guest interaction between (TBP)$_4$SiW and ditopic guest.** Because the cationic head locates at the end of one side arm, the four TBPs covering on SiW cluster still have free cavities for the inclusion of guest molecules. Thus, a linear molecule *para*-bis(4-cyanbutoxy) benzene (abbreviated as BCB) bearing two guest groups on both ends (Supplementary Figs. 35−37), which has strong binding capability to alkyl substituted pillar[5]arenes[36,37], is selected to bridge [TBP]$_4$SiW complex. By mixing (TBP)$_4$SiW and BCB with a molar ratio of 1:2 in chloroform, the host–guest inclusion can be detected. In comparison to the $^1$H NMR spectra of isolated host complex and guest molecule, obvious peak shifting and broadening of protons that are attributed to the inclusion groups and adjacent groups take place (Fig. 2). The upfield shifting of the protons belonging to BCB, sourcing from inclusion-induced shielding effect, and the 2D NOESY analysis between H(a−d) of host and H(2−5) of guest groups further illustrate the insertion of BCB into the cavity of TBP[38]. The titration experiment under a fixed BCB concentration (4.0 mM) suggests a slow exchange referring to NMR timescale (Supplementary Fig. 38). Upon addition of (TBP)$_4$SiW, the integral value of proton H(5) decreases gradually until the amount of (TBP)$_4$SiW reaches to the half of [BCB], revealing a stoichiometric inclusion ratio of 1:2 between (TBP)$_4$SiW and BCB. From a simplified binding model of TBP·Br and BCB, an identical 2:1 inclusion ratio is obtained (Supplementary Fig. 39) and the fitting titration curve of isothermal titration calorimetry (ITC) measurement (Supplementary Fig. 40) gives a total inclusion constant of $3.2(\pm 0.1) \times 10^6$ M$^{-2}$ in chloroform. The host–guest interaction

is sensitive to temperature and gets weak obviously near 60 °C (Supplementary Fig. 41). However, based on the fact that the disappeared signal of proton H(5) at room temperature does not reappear at raised temperature, the inclusion can be confirmed to maintain to some extent at higher temperature.

Such obtained inclusion system is extendible to the POM clusters bearing different chemical compositions with more or less surface charges, the cationic pillar[5]arane derivatives with different lengths of spacer, and other bola-form guest molecules with changed middle groups. The ionic complexation and inclusion regularity show consistent properties. The binding numbers of TBP analogs in the formed ionic complexes can be well controlled via both charge number and surface area limit of POMs to accommodate organic cations. The observed host–guest inclusion from TBP and its derivatives also suggest little relevance of their electrostatic interaction with the type of POMs, and the inclusion of the formed complex hosts to ditopic guests will become the decisive structural merit for the formation of supramolecular pseudorotaxane.

**Supramolecular gels and framework assembly structure.** Considering multiple binding sites on both host and guest units, their combination leads to a supramolecular polymerization[39]. In the case of four TBP groups surrounding SiW and two guest groups on BCB, a full inclusion will direct A$_4$B$_2$ type of connection with a high crosslinking degree. At the concentration of 0.08 mM or lower based on host complex, only particle co-assembly of (TBP)$_4$SiW and BCB is observed (Supplementary Fig. 42). With the concentration increasing gradually to a critical point of 0.15 mM, fibrous assembly appears and becomes full-grown at 0.25 mM with the assistance of sonication. Further increasing the concentration of the inclusion mixture results in the formation of supramolecular gels in chloroform and the inversion test gives the critical gelation concentration (CGC) 4.0 mM based on (TBP)$_4$SiW (1.85 wt%) (Supplementary Fig. 43). Significantly, after trying most of common solvents, only chloroform and its mixture with toluene are found to support the gelation of the inclusion complexes.

Since both isolated (TBP)$_4$SiW and BCB in chloroform are aggregated even at a low concentration (Supplementary Fig. 44), the full growth for crosslinked supramolecular polymerization requires an essential activation to break the original aggregation and then form the target assemblies. As a result, the sonication is applied to promote the dispersion of the two building units for the multiple host–guest inclusion-triggered crosslinking assembly while the defects yielding from the supramolecular polymerization due to the mismatched interaction can be greatly decreased in the dynamic process. Transmission electron microscopic (TEM) measurement discloses the morphologic evolution of assemblies vs. sonication time under a fixed concentration lower than CGC (Supplementary Fig. 45). The initial fibrous assemblies begin to appear after ca. 1.0 h of sonication and grow up to well-developed fibers with several tens of micrometers in length and some tens of nanometers in width within ca. 3.0 h. In the case of the concentration higher than CGC, the sustained sonication triggers the gelation and the well-developed fibrous assemblies with much larger size are observed within ca. 4.0 h (Fig. 3a). Further prolonging sonication time over 7.0 h does not show obvious sign that the gel is damaged. Regardless of above or below CGC, once the self-assembly fibers formed, they display a good stability against dilution, even down to several folds lower than the critical concentration at which fibrous self-assembly starts to emerge (Supplementary Fig. 46), revealing the superiority of multiple interactions in enhancing structural strength of the formed assemblies.

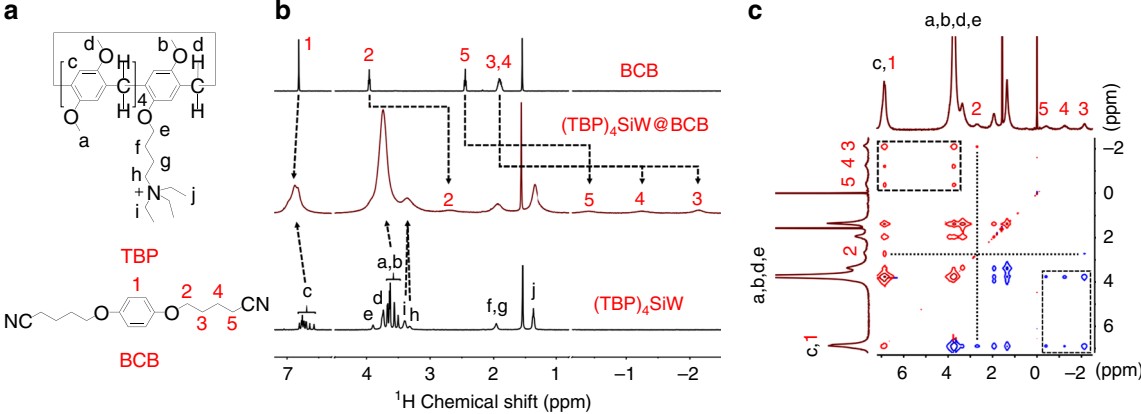

**Fig. 2 Host–guest interaction between ionic complex and BCB. a** Molecular structure of cationic host TBP and guest BCB. **b** Partial $^1$H NMR spectra (CDCl$_3$, 500 MHz, 25 °C) of BCB, (TBP)$_4$SiW, and (TBP)$_4$SiW@BCB (1:2). **c** Partial 2D NOESY spectrum (CDCl$_3$, 500 MHz, 25 °C) of (TBP)$_4$SiW@BCB (1:2). Source data are provided as a Source Data file.

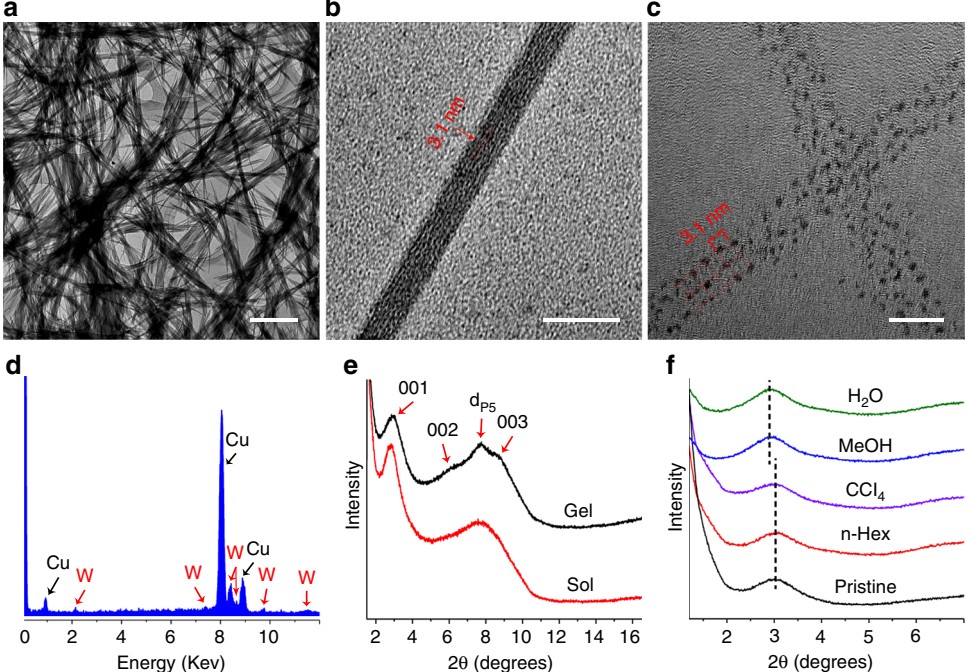

**Fig. 3 Co-assembly and supramolecular framework structure of (TBP)$_4$SiW@BCB in gel. a–c** TEM images of gel assemblies at different amplification scales. **d** Energy dispersive X-ray spectrum (EDX) focusing on the gel assembly. **e** Powder X-ray diffractions of sol and gel. **f** Powder X-ray diffractions of xerogel powder after treating with different solvents. Scale bars, 2 μm (**a**), 100 nm (**b**), 10 nm (**c**). Source data are provided as a Source Data file.

The detailed TEM image (Fig. 3b) illustrates a stripe-like self-assembled structure with an estimated spacing of ca. 3.1 nm along the focusing fibril. The high-resolution image for an exfoliated fibril shows that the dark spots in size of ca. 1.0 nm ascribing to SiW cluster distribute in a square like array with an average distance of ca. 3.1 nm (Fig. 3c), identical to the distance to the image under low amplification. EDX analysis (Fig. 3d) focusing on the fibers indicates the presence of tungsten element, which confirms the existence of inorganic clusters. With the square lattice distribution of SiW clusters, an orthotropic inclusion motif of host and guest components can be concluded. Apparently, in comparison to the tetrahedral connection fashion, such a planar framework binding style is more favorable for decreasing interfacial energy via a multilayer packing with the same components next to each other in the vertical direction. The reason is that the interfacial incompatibility deriving from the

exposure of hydrophilic SiW cluster in hydrophobic media can be diminished to the minimum through tight overlapping between top and bottom clusters locating at adjacent layers. On account of the diameter (1.0 nm) of SiW cluster, the height (0.72 nm) of the methoxy pillar[5]arene[40], the length (1.8 nm) of BCB, and the host–guest recognition mode (Supplementary Fig. 47), the calculated distance of each side for an orthotropic lattice is about 3.2 nm. Considering the possible twisting of cationic head, neck spacer as well as tilting of the guest molecule, the value estimated from TEM image is very close to the calculated length. Thus, it can be concluded that a tight packed two-dimensional framework structure possessing tetragonal pores builds the fibrous assemblies, leading to supramolecular gels (Fig. 1a). The multi-order diffractions (Fig. 3e, Supplementary Fig. 48) provide a $d$ value of 3.0 nm, just corresponding to the distance between two clusters, further supporting the assignment of tetragonal framework

assembly. In addition, the distribution peak obtained by Non-Local Density Functional Theory (NLDFT) method in a $N_2$ sorption experiment at 77 K gives a pore size of ca. 1.9 nm (Supplementary Fig. 49), which is in perfect consistence with the calculated inner pore diameter (ca. 1.8–2.0 nm) according to above structure speculation, further supporting the proposed tetragonal framework structure in gel assemblies. A macro-pore size with a distribution over 150 nm, which is obtained from the BET method, can be ascribed to the random stacking of the gel fibers.

Rheology experiments are carried out to examine the viscoelasticity behavior. The formed gel maintains the state within several months under a sealed condition below 50 °C. The dynamic storage modulus (G') and loss modulus (G") vs. the angular frequency (ω) at a strain amplitude of 0.5% are recorded at the concentration of 10.0 mM. The G' values maintain constantly higher than G" values within the entire frequency range (Supplementary Fig. 50a), indicating a typical gel character. The gel nature retains over a wide region of strain amplitudes. For the gels comprising of other POMs bearing four negative charges, their G' values remain at a similar level (Supplementary Fig. 50b, c). However, the G' value of (TBP)$_4$SiW@BCB is higher than those of reported pillar[n]arene-based supramolecular gels[41,42], while the gels constructed by other host complexes with prolonged flexible spacers in TBP show decreased G' values (Supplementary Fig. 50d−f), suggesting the important roles of both framework structure and complex rigidity for gel's stiffness. At a cyclic oscillation strain amplitude of 100%, the gels transform into sol state with high mobility, yet once the strain amplitude is released to 0.5%, the G' values get back to the original range and become higher than G" again (Supplementary Fig. 51). This process can be repeated over ten times without apparent loss of modulus, indicative of a rapid self-healing property of the gels against straining[43], as confirmed in the following membrane processing.

## Gel membrane separation of immiscible liquids and consecutive in situ switching

The gel membrane is prepared following a simple spreading procedure of a diluted supramolecular gel. Typically, the gel coatings, regardless of surface shape, size, and curvature of the substrate, are obtained by stepwise spin coating by controlling the speed lower than 2000 r s$^{-1}$ or just a dip coating. The support can be various materials such as stainless-steel mesh, filter paper, silicon wafer, glass slide, etc. The membrane thickness is adjusted by controlling either gel concentration or coating volume. Scanning electronic microscopic (SEM) characterization validates that the surface of substrates has been covered evenly with gel membrane without obvious defects (Fig. 4a–c). The lateral observation demonstrates the gel membrane possessing an average thickness over one hundred nanometers at the condition of separation (Supplementary Fig. 52). The prepared gel membrane performs a flexible feature during processing in the presence of chloroform. By fixing the membrane that spreads on a steel mesh to the middle of two glass tubes fastening with a screw clamp, a separation device is obtained. The possible defects and cracks during the preparation of membrane device can be well healed by a simple wetting treatment of gelation solvent chloroform.

The membrane performs high stability in pure water, saturated sodium chloride, acidic and alkaline aqueous solutions within pH 1–12 (Supplementary Fig. 53), alcohols, and non-polar as well as partial weak polar organic solvents, whereas it becomes instable in some strong polar that are miscible with water and polar organic solvents that are immiscible with water (Supplementary Tables 1, 2). Because the electrostatic, host–guest inclusion, and interlayer interactions are the main driving forces of the supramolecular framework fibers, their combination strength is closely related to the membrane structure. Apparently, the electrostatic interaction is not affected by the liquid environments in the present study because of the membrane retains well in

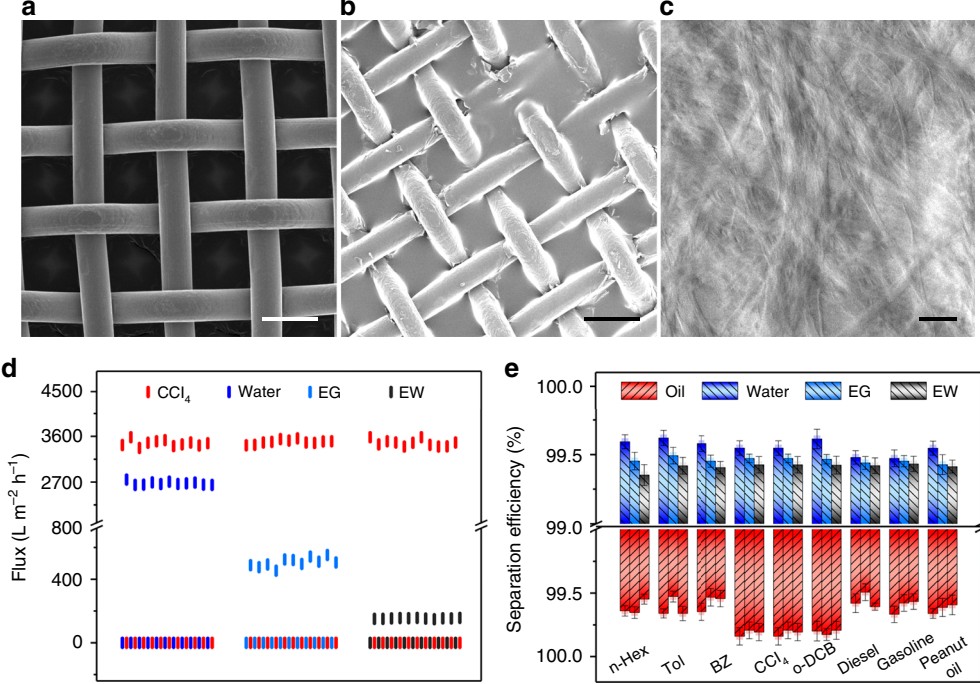

**Fig. 4 Characterization of (TBP)$_4$SiW@BCB gel membrane and separation property. a–c** SEM images of (**a**) blank stainless-steel mesh, (**b**) gel membrane coating, and (**c**) local amplification. **d–e** column plots of (**d**) flux plots and (**e**) separation efficiency of some immiscible liquid pairs encountering over 10 reversible transformation cycles under gravity, where EG: ethylene glycol, GW: glycerin water solution (67.7%), n-Hex: n-hexane, o-DCB: o-dichlorobenzene, Food oil: peanut oil. Data are collected to get average values (n = 5). Scale bars, 50 μm (**a, b**), 2 μm (**c**). Source data are provided as a Source Data file.

aqueous solutions with strong electrolytes and strong polar organic media (Supplementary Fig. 54). The interlayer interaction between framework layers comprising of both hydrophobic and hydrophilic components is in principle sensitive to the polarity of organic solvents such as strong polar solvent DMSO, which could not break the electrostatic and host–guest interactions but shows dissolution to the membrane, implying its weakening to the interlayer interaction. In contrast, a few other solvents including DMF with similar polarity to DMSO display dissociation effect for the host–guest interaction (Supplementary Fig. 55). Other examples reveal that the breaking to the host–guest interaction can be ascribed to either solvent polarity or the competitive inclusion. These results imply that both molecular structure and solvent polarity may lead to the membrane unavailable. In general, the disruptive effect of organic liquids can be eliminated by mixing water up to 70% for water miscible solvents and greatly reducing polarity for water-immiscible solvents. With the suitable improvement, the supramolecular gel membrane can be used for a larger scope of separation of immiscible organic liquids and water with enough structural stability[44].

Considering the initial hydrophobicity of the membrane, hydrophobic liquids such as carbon tetrachloride and hexane, are firstly performed for permeation test, and they pass through very quickly under gravity. In contrary to hydrophobic liquids, water, ethylene glycol, and aqueous glycerol do not flow past the membrane in a separated experiment. In the case of mixture liquid of carbon tetrachloride and water, the hydrophobic part passes through the membrane while water is blocked, and thus, an automatic separation of oily liquid and water is realized. Series water-immiscible oils from water and immiscible oily liquids such as toluene, benzene, dichlorobenzene, diesel, gasoline, and peanut oil, have been proved to be separable via the membrane (Supplementary Table 2). For the liquids that have lower specific gravity values than water, such as benzene and hexane, they are separated easily from water by inclining the device to make those low-density liquids touch the membrane. Beside oil-water system, immiscible organic liquid mixtures such as carbon tetrachloride and ethylene glycol, carbon tetrachloride and glycerin aqueous solution (67%), are also proved to be applicable for direct separation through the supramolecular gel membrane. Interestingly, the surface property of separation membrane can be changed by using a joystick liquid, such as methanol, ethanol, isopropanol, and tetrahydrofuran, which do not damage the assembly structure but are miscible with both hydrophobic and hydrophilic liquids to be separated. Typically, in the case of hydrophobic state, the addition of methanol drives pre-blocked water to go through the membrane and a further wetting with water makes the membrane no longer wettable by hydrophobic liquids. As a result, water passes through the membrane automatically while oily liquids become blocked. More importantly, the changing over of the separation membrane back to the initial hydrophobic state is achieved following the same route by using joystick liquids. For instance, when methanol is added to the hydrophobic liquids blocking on the membrane device, accompanying by the mixture flowing out, the hydrophilic membrane becomes wettable for the blocked liquid. Further wetting with the hydrophobic liquid makes the membrane hydrophobic and only the hydrophobic liquids immiscible with water are passable while water is blocked again to accomplish a conversion cycle (Supplementary Video 1). Such a separation switching process is facilely in-situ operated in the presence of blocked liquid and can be repeated at least twenty times (Fig. 4d) without obvious flux loss, indicating the high stability of membrane and the resistance to the liquid fouling. A consecutive separation membrane under wetting state performs a capability on intermittent separation. For a used membrane having dried for a few days, the separation feature can be quickly recovered completely by a simple wetting with chloroform, while the separation efficiency and flux maintains almost the same level as that at the initial state (Supplementary Fig. 56).

The separation efficiency of both hydrophobic and hydrophilic liquids reaches over 99% for all the measured mixtures (Fig. 4e). The hydrophobic membrane with a fixed membrane thickness ($182 \pm 20$ nm) performs an average flux about 3500 L m$^{-2}$ h$^{-1}$ for oily liquids under gravity (Supplementary Fig. 57) and the flux decreases to ca. 3000 L m$^{-2}$ h$^{-1}$ with the thickness increasing to ca. 380 nm. For the membrane at hydrophilic state, the flux for water decreases to about 2700 L m$^{-2}$ h$^{-1}$, and becomes much lower for hydrophilic liquids with lower fluidity. The increased viscosity does not affect separation efficiency of liquids obviously but leads to a larger decrease of flux of the liquids to be separated.

**Mechanism discussion and simulation.** Static contact angle (CA) experiment is carried out to understand the oil-water separation behavior and wettability modulation of the gel membranes (Fig. 5a, c). The CA value of initial gel membrane for water is ca. $165 \pm 3°$ under oil, indicative of a superhydrophobic surface. For the membrane undergoing a treatment of methanol and then

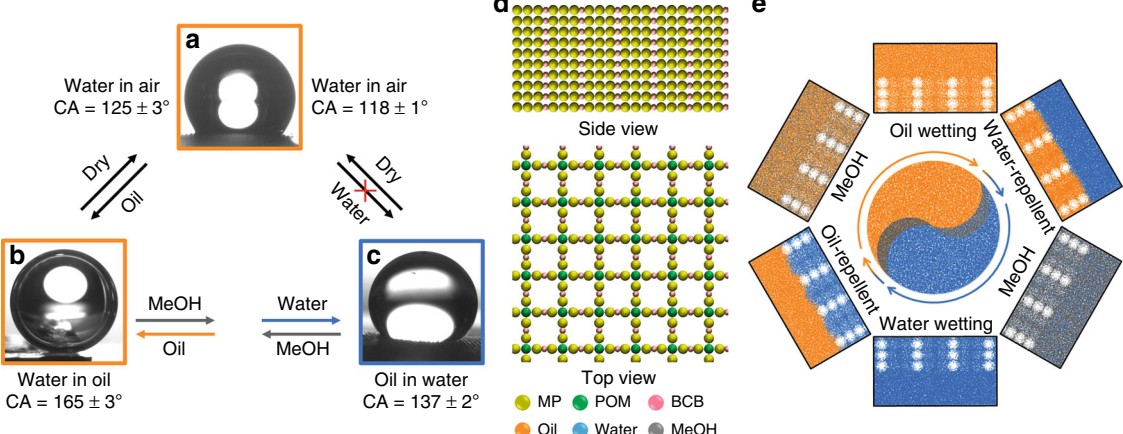

**Fig. 5 Surface property and DPD simulation of separation process. a–c** Static contact angle changes from (**a**) dried framework membrane through (**b**) wetting with chloroform to (**c**) the one washing with methanol and water, and then getting back to dried state, data are collected to get average values ($n = 5$). **d–e** The DPD packing model in (**d**) top and side views, in which the chemical species are extracted as particles with corresponding properties, where MP represents methyl-pillar[5]arene and the oil denotes non-polar liquids like carbon tetrachloride. Source data are provided as a Source Data file.

water, the CA value for oil reaches ca. $137 \pm 2°$ under water, showing a highly oleophobic surface. Though the surface does not perform superoleophobicity, it is applicable for intercepting oily liquids under a hydrophilic environment[45]. As both hydrophobic and hydrophilic components in the framework assemblies distribute evenly, the enhanced surface properties of the separation membrane can be concluded to source from an integrated contribution of assembly structure and surface morphology. On one hand, the rough surface of membrane comprising of supramolecular nano-fibers (Supplementary Fig. 58) is favorable for enhancing both hydrophobicity and hydrophilicity[46,47]. On the other hand, the nanosized pores in supramolecular frameworks play a critical role in boosting up the surface property based on the model like Nepenthes pitcher plants and fish scales[48,49]. In the case of wetting with carbon tetrachloride, the hydrophobic organic components in framework are compatible at an outspread state, which shields the hydrophilic cluster and thus allows hydrophobic liquid to fill in the inner nano-channels within framework assemblies. Apparently, the hydrophobic liquid wetted surface strengthens the hydrophobicity of separation membrane because the CA value of water for the same membrane at dried state is only $125 \pm 3°$. In the case of wetting with water, the hydrophilic SiW cluster becomes compatible and tends to be exposed to a polar environment while the hydrophobic inclusion components tend to avoid the polar environment surrounding POM, and therefore, the increased hydrophilicity allows hydrophilic liquid to fill in the framework nanopores. As a result, the CA value of oil becomes much larger than the membrane in dried status, confirming that water wetting results in a surface property transformation and a large enhancement of surface hydrophobicity. According to the model based on the minimization of system's free energy (Supplementary Fig. 59), the estimated free energy change ($\Delta E$) and intrusion pressure analysis ($\Delta P$) from series contact angle measurements in air and under liquids further point out the feasibility of immiscible liquid separation in principle (Supplementary Tables 3, 4). The larger energy change values under oily liquids than in the air figure out the infusion effect of miscible liquids into the framework structure for the separation of immiscible liquids. The adaptable effect of the supramolecular framework to the external environment polarity is also supported by the size change of framework lattice (Fig. 3f) though the scale is very small. After wetting the membrane with polar solvents such as methanol and water, the statistical analysis for side length calculated from primary diffraction (001) in dried xerogel shows an increase of ca. $1.0 \text{ Å}$, in comparison to the pristine state or wetting by carbon tetrachloride and n-hexane. As an indirect evidence, the CA value of the membrane wetted by water is smaller than that wetted by non-polar solvent at dried state, revealing the elastic behavior occurring at the framework vs. the polarity change.

With the proposed model that the framework assembly in gel fibers promotes the separation of oils and water by alternate wetting with hydrophobic or hydrophilic liquid, continuous switching during separation process can be explained rationally. However, it should be noticed that the above oil or water switching process does not occur automatically without the assistance of intermediate state. Because the joystick liquid methanol is miscible with both oily carbon tetrachloride and water, its rinsing to the membrane results in each of the liquids that previously fill in the nanopores to be replaced. Thus, the methanol-filled surface of fibrous assemblies in the membrane becomes amphiphilic and can be further substituted with other miscible liquids, regardless of hydrophobic or hydrophilic. The compatible environment from methanol allows the membrane to turn into either hydrophobic or hydrophilic surface depending on the subsequent wetting liquid.

In order to account for the joystick liquid-jointed switching process bearing the essential feature of framework structure, a particle architecture bearing a composition of $[(TBP)_4SiW\cdot(BCB)_2]_n$ is used in the dissipative particle dynamic (DPD) simulation (Fig. 5d, Supplementary Fig. 60). The canonical ensemble simulations are performed in a three-dimensional cubic box of size $46^3$ with periodic boundaries[50]. The interaction change between particles with interaction parameters $\alpha_{ij}$ (Supplementary Table 5) vs. time is set to obey Newton's motion equations[51]. Upon placing a framework model into $CCl_4$ media, the orange particles ($CCl_4$) immediately spread into the porous framework, leading to a continuous phase (Fig. 5e top). This implies that the framework displays an attractive feature with regard to $CCl_4$ particles. When applying a mixture of $CCl_4$ and water particles into the model, only orange particles move close and spread into the model framework while blue particles (water) tend to be inaccessible (Fig. 5e top right). The result shows consistency with the fact that $CCl_4$ flows past gel membrane in the case of superhydrophobic wettability under oil. Since methanol mixes with $CCl_4$ and water at any volume ratio, it is treated to be attractive for the two kinds of particles. When mixing with $CCl_4$, the gray particles defined as methanol firstly replace those orange particles that locate inside the framework assembly (Fig. 5e bottom right), and then the blue particles representing water start to fill in and finally purge gray particles out thoroughly accompanying by a succedent dynamic diffusion process (Fig. 5e bottom). Meanwhile, one can see that the framework becomes accessible to blue particles but repellent to orange particle (Fig. 5e bottom left). Definitely, since the gray particles also replace blue particles in the framework structure (Fig. 5e top left), a reverse process for the framework to be occupied by orange particles also takes place by placing the system in a hydrophobic environment.

## Discussion

We have shown a strategy to fabricate flexible supramolecular framework self-assemblies. In the formed two-dimensional porous structure, electrostatic interaction drives an inorganic polyanionic cluster acting as a node to connect four organic cationic pillar[5]arene molecules serving as a host to dominate the inclusion of the formed host complex to the bridging nonionic guest at a preferable direction. The four charges of inorganic cluster decide the orthogonal binding positions within the framework structure and the hydrophobic/hydrophilic property of inorganic and organic components strengthens the layered packing of the framework assembly, forming porous channels perpendicular to the layer surface. In contrast to well-known frameworks which are constructed via coordination or covalent bond, the intermolecular interactions in the present research not only provide enough structural rigidity and uniform porosity, but also bring about binding flexibility and processability. Such unique characteristics of supramolecular framework structure enlighten the important potentials of flexible nanoporous materials. The gelation in weak polar organic environment allows the layered framework assemblies to be facilely processed into separation membranes through a simple coating procedure. The synergistic effect of the formed framework structure and the surface characteristic of components paves the membrane for quick separation of oil-water and immiscible liquids as well as the in-situ changing over for the separated liquids reversibly. The observed response behaviors of the separation membrane also demonstrate the advantage of flexible frameworks. It could be expected that such a kind of supramolecular framework assembly can be used for selective adsorption and separation of chemical objects such as polypeptides and proteins with hydrophilic or hydrophobic surface. Based on the same strategy, the property on

solid surface coating with the framework assembly can be modulated conveniently.

## Methods

**Ionic complexes**. To a solution of cationic pillar[5]arene in methanol or methanol/ $H_2O$ (20 mL), POMs in methanol or methanol/ $H_2O$ (20 mL) was added slowly with vigorous stirring at room temperature. After 2 h of ionic exchange reaction, the formed precipitate was filtered and washed with deionized water (30 mL × 3) and then methanol (30 mL × 3), dried under vacuum, giving ionic complex. See supplementary information for a detailed Methods section.

**Supramolecular gel and fibrous assemblies**. $(TBP)_4SiW$ (19.34 mg, 3 mmol) and BCB (1.63 mg, 6 mmol) at 2:1 molar ratio was mixed in 500 μL of chloroform. After 4 h sonication of the solution the supramolecular gel formed for structural measurements and post-treatment. The xerogel powder dissolves in dimethyl sulfoxide (DMSO), dimethyl formamide (DMF), formamide, and dichloromethane, but not in water, ethylene glycol, glycerol, methanol, ethanol, acetonitrile, acetone, tetrachloromethane, toluene, and hexane et. al. The long fibers were prepared by using the same procedure within a shorter sonication time, such as 1−3 h before the formation of gels. For other gel samples prepared from others supramolecular hosts, $(TBP)_4PWV$, $(TBP)_4PMoV$, $(THP)_4SiW$, $(TOP)_4SiW$, and $(TDP)_4SiW$ were used following the same methods under different concentrations.

**Framework assembly gel membrane**. The gel membranes were prepared through a simple spin-coating procedure. The chloroform solution for spin coating was prepared from diluting above prepared supramolecular gel or the pre-gelation solution after sonicating 3 h. Commercial solid substrates, stainless-steel mesh, nonwoven, filter paper, silicon wafer, and copper grid, were used depending on the purpose for various measurements or separations. The spinning speeds were set at $400\ r\ s^{-1}$ for 30 s and then $2000\ r\ s^{-1}$ for 60 s. The concentration was normally 1.00 mM based on $(TBP)_4SiW$ (if not specified). After dryness, the prepared membranes were applied for measurements. Here, some changed preparation conditions differing from that of the membrane for separation were made for getting a clear structural observation depending on the instruments.

**Samples for TEM measurement**. To obtain a higher resolution image of the gel nanoparticles, a dilute gel solution was used for TEM measurement, because concentration in gel state is difficult to get clear images. The chloroform solutions with concentrations of 0.25, 0.05, 0.01 mM, based on $(TBP)_4SiW$, was used. Here, we used a thin copper ring to capture a thin membrane and then cast onto a copper grid at once. During the measurement, high-energy electron beam was used to sweep the surface of assemblies in an instant repeatedly to get an image with clear contrast. Smart camera technique was used for the image collection. The position and size of clusters showed a slight distortion because of the ghosting phenomenon of image superposition.

For observation of the gelation vs. sonication time, we fixed the concentration at 0.25 mM based on $(TBP)_4SiW$. During the observation for the formation of fibers vs. the sonication, samples were taken every 1.0 h.

**Samples for rheology measurement**. The gels used for rheology measurement were prepared at a concentration of 10 mM based on ionic complexes. The gap distance between two 25 mm ETC stainless parallel plates was 0.2 mm and all experiments were carried out at temperature of 25 °C. Oscillatory frequency sweeps were set in the range from 0.1 to 100 rad $s^{-1}$ at an oscillation amplitude (g = 0.5%). The cyclic experiments were conducted under a continuous angular frequency (ω = 10.0 rad $s^{-1}$) with alternating oscillation amplitude between 0.5 and 100%.

**Separation efficiency (η) calculation:**

$$\eta = \frac{m}{m_0} \times 100\%$$

where $m$ and $m_0$ are the weight of liquids to be separated before and after the separation. The possible adsorption difference was ignored, because pre-wetting to the membrane was carried out before the separation experiments. The last five parallel experiments were adopted to get average values.

**Flux (J) calculation under gravity:**

$$J = \frac{V}{t \times A}$$

where $V$ is the filtrate volume (in L), $A$ is the effective filtration membrane area (in $m^2$), and $t$ is the separation time (h). The data were collected from the average of five parallel experiments.

See supplementary information for a detailed Methods section.

## Data availability

All data supporting the findings of this study are included in this article and the supplementary information and are also available from the corresponding author upon request. The source data are provided as a Source Data file.

The source data underlying Figs. 2−5 and Supplementary Figs. 1−60 are provided as a Source Data file.

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

## Acknowledgements

The authors acknowledge the financial supports received from the National Natural Science Foundation of China (21574057 and 21833008), the Changbaishan Distinguished Professor Funding of Jilin Province, and the Program for JLU Science and Technology Innovative Research Team (2017TD-10). The authors thank Chao Xiao from Sichuan University for kindly help in ITC measurement.

## Author contributions

L.X.W. designed and guided the project. G.H.Z. carried out synthesis and structural characterizations. X.F.C. helped in partial synthesis. B.L. performed XRD experiment and analysis. B.Y.L. and Z.Y.L. finished the computational investigation and provided theoretical analysis. Y.Z. participated in data corrections. L.X.W. and G.H.Z. finished the preparation of final manuscript.

## Competing interests

The authors declare no competing interests.
