## [Peer Review File · Nature Communications]

Editorial Note: Parts of this Peer Review File have been redacted as indicated to maintain the confidentiality of unpublished data

Reviewers' comments:

Reviewer #1 (Remarks to the Author):

In this paper, authors developed a kind of supramolecular frameworks that are composed of inorganic polyanionic cluster and cationic organic host as well as bridging guest. The supramolecular coated membrane achieved the separation of oil-water by adjusting the surface hydrophobic and hydrophilic properties by H₂O and methanol/ethanol/tetrahydrofuran. However, neither the structure of the constructed membrane nor the oil-water separation was that significant. Besides, the language of this paper was so roughly written that spelling and grammatical mistakes made it difficult to read. In my opinion, this paper doesn't fulfill the quality to be published in Nature Communications, and my suggestion is rejection. My major concerns regarding this paper are listed below.

1. The author described that the membrane could intrinsically resist the weak polar and nonpolar organic media; however, some other organic solvents which were immiscible with water can damage the membrane due to their dissolution to the assembly and breaking inclusion interaction. Here I'm curious why some organic solvents can dissolve the membrane while some others would not? Is this due to some functional groups of organic solvent that led to this result?

2. The novelty of this work is not high enough, for it's natural that membranes with hydrophobic surface can stop water from passing through while allowing the nonpolar organic solvent flow past it; and similarly, membranes with hydrophilic surfaces will surely allow water to pass through while blocking the nonpolar organic solvent from passing through it.

3. From the video, as well as the description in page 6 of the paper, two kinds of immiscible liquid, for example, H₂O (blue) and CCl₄ (red), could be separated by adjusting the hydrophilicity and hydrophobicity of the membrane with H₂O and methanol, alternately. However, I'm afraid that this separation is not that significant because the two species of the liquids are intrinsically immiscible and consequently located in a different layer. I believe it will be more significant if this membrane could manage to separate the homogeneous mixture of oil and water.

4. The author claimed the separation efficiency of both hydrophobic and hydrophilic liquids could reach over 99% for all measured mixtures. How was this datum of the separation efficiency experimentally obtained? Did the authors weigh the solvents after passing through the membrane, or by some method else? No detailed information on this part of the experiment was given in the paper.

5. What is the effect of membrane fouled after each use?

6. With regards to the proposed mechanism, the author did nothing but static contact angle, which is absolutely not convincing.

7. This paper is very badly written, and too many spelling mistakes and syntax errors appeared, for example (the errors listed as follows are only a small part, and there are more writing mistakes throughout the paper):

(1) ...exhibit vigorous materials potential applications...("vigorous" in Line 1 in "Abstract");

(2) ...processable conveniently because of their normal state...("because" in Line 3 in "Abstract");

(3) ...could be a preferable approach for structural flexibility...("preferable" in Line 5 in "Abstract");

(4) Ordered nanoporous structure ...exhibit...(Line 1 in "Abstract", the subject word "structure" should be plural form.);

(5) ...during separation of immiscible liquids...(Line 9 in "Abstract", it should be "...during the separation of immiscible liquids..."). Too many such kinds of mistakes appeared in the whole paper.

(6) ...structural symmetry, play a key role for diverse fantastic frameworks...(Line 3 in paragraph 1, "play a key role for..." should be "play a key role in....")

(7) ...materials with suitable solvent trapping ability is an alternative manner...(Line 1 in paragraph 2, should be "are" instead of "is" because the subject word is the plural form)

Reviewer #2 (Remarks to the Author):

In this manuscript, the authors presented flexible supramolecular frameworks which are composed of inorganic polyanionic cluster, cationic organic host, and bridging guest. In the frameworks, the inorganic polyanionic cluster is connected with four organic cationic pillar[5]arene molecules by electrostatic interaction. And the organic cationic molecules are further used as a host to inclusion of the formed host complex with bridging nonionic guest. The resulting framework structures are able to form layered nanofibrous assemblies with enough structural rigidity and uniform porosity. Moreover, the framework structures could form supramolecular gels, which are favorable for processability. Interestingly, the surface properties of the porous frameworks can be modulated by filling hydrophilic/hydrophobic liquids, which renders the frameworks can be employed as a switchable oil-water separation membrane. The works are interesting and important, and thus I recommend the acceptance of the manuscript after addressing the following minor issues.

1. The authors should check all the Figure numbers. For example, the "Ionic complex (TBP)₄SiW" section at page 6 in the Electronic Supplementary Information, the "Supplementary Figure 16" should be changed to "Supplementary Figure 17", and the original "Supplementary Figure 17" should be changed to "Supplementary Figure 18".

2. In the application section, the methanol was used as joystick liquids. I want to know whether other solvents can be used as joystick liquids, such as DMF, THF, and DMSO.

Reviewer #3 (Remarks to the Author):

The manuscript describes the remarkable formation of a 2D, porous, supramolecular self-assembly made of hydrophilic, fully-oxidized Keggin-type POMs bearing the 4- charge and hydrophobic organic ligands for control of bonding and bridging between the assembled units. Contrary to the frameworks generated by coordinative or covalent bonds, the framework structure produced by the authors of this work is benefited from intermolecular interactions. The compounds are fully characterized and the results reported are adequately supported by the acquired data from different well-executed measurements. The applicability of organic-inorganic, POM-based materials in gel membranes for the controlled switching of oil-water separation has been demonstrated.

In my opinion, this complete work deserves to be published in Nature Commun. after minor revision.

The authors should comment in the main text on:

(1) if the described self-assembly strategy can be extended to any other POM family, like e.g. Lindqvist, Anderson or Wells-Dawson structures?

(2) what is the highest POM-charge that still remains feasible for the construction of such supramolecular structures?

(3) what role does the Keggin-POM isomerism play for the framework assembly and its stability?

Literature to be quoted:

(1) Polyoxometalates as components of supramolecular assemblies. Chem. Sci. 2019, 10, 4364-4376.

(2) Interplaying the amphipathic polyoxometalate interactions in solution and at solid-liquid interfaces: A toolbox for the technical application. *Nanoscale* 2019, 11, 4267-4277.

Typos:

(1) p.2: MOLDI-TOF  MALDI-TOF

(2) p.3: A reference doesn't appear for the sentence ending with "... the insertion of BCB into the cavity of TBP."

Responses to Reviewers' comments

Reply for Reviewer 1

General Comments: In this paper, authors developed a kind of supramolecular frameworks that are composed of inorganic polyanionic cluster and cationic organic host as well as bridging guest. The supramolecular coated membrane achieved the separation of oil-water by adjusting the surface hydrophobic and hydrophilic properties by H₂O and methanol/ethanol/tetrahydrofuran. However, neither the structure of the constructed membrane nor the oil-water separation was that significant. Besides, the language of this paper was so roughly written that spelling and grammatical mistakes made it difficult to read. In my opinion, this paper doesn't fulfill the quality to be published in Nature Communications, and my suggestion is rejection.

Reply: We thank the comments on our manuscript and we appreciated to have an opportunity to address more about motivation to the present research results. For the conclusive comment "neither the structure of the constructed membrane nor the oil-water separation was that significant", we do not think it was true, especially in the case of no explicit explanations given to support such an assertive view. Regarding the membrane structure, though made of fibers like many of oil-water separation membranes, it was actually different from all those in published results. The reason is that, on one hand, the fiber comprising the membrane is automatically formed from the self-supporting gel assembly of small molecular components through intermolecular interaction, while on the

other hand, the obtained fibers represent a kind of flexible supramolecular framework structure that differs completely from both crystalline metal organic framework (MOF) and disordered porous polymers but bear the common advantages of them. As for the building unit and driving force, only this group reported the inorganic cluster nodes and the combination of ionic interaction and host-guest inclusion. For the oily liquids-water separation process, we realized the membrane application of the flexible supramolecular framework fibers and the in-situ consecutive switching of oil and water as well as immiscible organic liquids, based on the amphiphilicity of different molecular components locating at the framework. Not limited to above, in comparison to those reported works, such flexible framework structure fibers and membranes display large expansibility in extensive potentials. For example, we just used the same membrane composite with general polymer to separate soft microemulsions (both O/W and W/O) with droplet size as small as ca. 2 nm as well as rigid metal nanoparticles with the same size scale. These novel properties on the aspects of self-assembled structure and membrane application show unique importance both in fundamental research and supramolecular materials. For the English spelling and expressions in writing, we carried out a thorough polishing to make it more readable. To emphasize the importance of oil-water separation and the application of membranes, we added a short paragraph marked in yellow background in the first page.

Question 1: The author described that the membrane could intrinsically resist the weak polar and nonpolar organic media; however, some other organic solvents which were immiscible with water can damage the membrane due to their dissolution to the assembly

and breaking inclusion interaction. Here I'm curious why some organic solvents can dissolve the membrane while some others would not? Is this due to some functional groups of organic solvent that led to this result?

Reply : Thanks for the question. The solubility of normal polymers is mainly related to their chemical structure and the solvents used. Highly crosslinked polymers could not be dissolved but swelled normally. As a kind of supramolecular polymers, the formed framework fiber (highly crosslinked) membranes can be dissolved by some solvents via breaking the assembly structure and the driving force due to the specific property of intermolecular interaction. Ionic interaction, host-guest inclusion, and interlayer interaction in the present study are three main driving forces that can trigger the dissolution of the separation membrane via breaking the self-assembled structure. In the step to prepare the ionic complexes (yielded from deposition in methanol), we have demonstrated that the ionic interaction is stable in all solvents used in the present work, even in the presence of inorganic salt aqueous solution. Therefore, the ionic interaction should not be dissociated by those solvents. The host-guest interaction is well known to be pretty sensitive to the solvent environments (J. Phys. Chem. B, 2015, 119, 6711–6720). Decreased polarity is favorable to stabilize the host-guest interaction while the increased polarity is unfavorable for such an interaction in the inclusion system of pillararenes. Because of the insolubility of isolated components, however, the fibrous assembly also shows stability in strong polar organic solvents such as those mixing with water since the insoluble packing structure helps to block the dissociation of host-guest interaction in the membrane. In contrast to this, the host-guest interaction in other polar solvents that is not miscible with water,

including those summarized in ESI Table 1 with complemented data, was reported to become weakened greatly, due to the decreased dipolar interaction and competitive interaction of solvent molecules, as mentioned by this Reviewer. The complemented ^1H NMR spectrum in deuterated dichloromethane, which is added as ESI Figure 59, shows the decreased host-guest inclusion, supporting above explanation. The damage from other special solvents like DMSO for the membrane can be attributed to a similar reason in breaking the host-guest interaction. As for the influence of solvents on the interlayer interaction, we believe that the mentioned solvents cannot break the layered packing because interfacial energy is unfavorable for the POM domain exposing to the solvent environment.

Question 2: The novelty of this work is not high enough, for it's natural that membranes with hydrophobic surface can stop water from passing through while allowing the nonpolar organic solvent flow past it; and similarly, membranes with hydrophilic surfaces will surely allow water to pass through while blocking the nonpolar organic solvent from passing through it.

Reply: We do not agree with the comment. As described in the main text and highlighted in cover letter, several important results on both material structure and the separation of oil-water with reversible switching property as well as the expansibility in applications have demonstrated the novelty of the present work. Firstly, the new hybrid organic gels are constructed from the supramolecular framework fiber in comparing to most reported organic and polymer gels that have general solid packing structures; Secondly, the processable organic gel can be facilely used for self-standing separation membrane with

self-healing property and stability in many solvents; Thirdly, relying on the unique framework fiber structure, the in-situ consecutive reversible switches of oil and water are realized. From material point of view, very few publications dealt with the flexible supramolecular frameworks and no publications concerned the fabrication of small molecular gels for oil-water separation membrane as far as we know. It is just the unique structural properties sourcing from the supramolecular strategy to make the membrane here display reversibility and self-healing behaviors, which traditional polymers and polymer assemblies are hard to perform naturally. From the oil-water separation point of view, switchable liquid-liquid separations were of course reported, yet a consecutive switching for liquid-liquid separation via a gravity method is still unknown up to date and our work disclosed such a convenient process, which is very much significant in practical systems.

On the other hand, the argument is also not suitable. The hydrophobic membrane allows the nonpolar organic solvent to flow past but it does not mean that it can stop water automatically. Similarly, the hydrophilic membrane follows a similar principle. The reason is that based on Laplace formula (Macromolecules, 1999, 32, 6800–6806), water can pass through a hydrophobic membrane when the gravity of water is higher than its infiltration pressure that depends on the contact angle and pore's scale in the membranes (Adv. Mater. 2011, 23, 4270–4273; Nat. Commun., 2012, 3, 1025; J. Mater. Chem., 2012, 22, 19652–19657; Langmuir, 2015, 31, 1393–1399; Adv. Mater. Interfaces, 2016, 3, 1600461).

Different from those traditional separation approaches that experienced chemical and physical modification, immiscible liquid modulation for surface property of separation membranes represents a new strategy (Nature, 2011, 477, 443–447; Adv. Mater., 2011, 23, 4270–4273; Nature, 2015, 519, 70–73; Nat. Commun., 2017, 8, 575; Nat. Rev. Mater., 2017, 2, 17036). However, all those published methods have encountered complicate process during membrane preparation and the operations for reversible switching need multi-steps. In contrast to this, the gel membrane with framework fibers not only has the intrinsic feature to hold organic liquids and water but also can be washed off through the intermediate solvents. We believed that it was a new contribution to the smart membrane separations.

Question 3: From the video, as well as the description in page 6 of the paper, two kinds of immiscible liquid, for example, H₂O (blue) and CCl₄ (red), could be separated by adjusting the hydrophilicity and hydrophobicity of the membrane with H₂O and methanol, alternately. However, I'm afraid that this separation is not that significant because the two species of the liquids are intrinsically immiscible and consequently located in a different layer. I believe it will be more significant if this membrane could manage to separate the homogeneous mixture of oil and water.

Reply: The comment seemed correct but it was not true. In many actual systems, oil and water do not exist in only two layers with a clear interface. Instead of that, the bulk oil and water mixtures are often cut into a great many small biphasic domains which cannot be separated as easily as ideal two phases that the comment considered. Based on the dispersion states of oily components and their density, viscosity, and chemical

composition, several known methods such as gravitational-/centrifugal- separation, electro-separation, demulsification, flocculence, and floatation separation, are usually used. However, these traditional routes cause low efficiency, high energy consumption, and often require the addition of extra chemicals. Thus, a strategy to simplify the separation process at a lower cost is highly desired. As a simple model to evaluate a new membrane material, the biphasic is favorable for excluding the influence of other misleading factors. As indicated in the main text, several oily liquid-water compositions including immiscible organic liquids were confirmed to be applicable for the separation of actual systems. In addition, we would like to remind that only methanol and similar organic solvents were used as the joystick solvents, not water.

We agree with the argument “it will be more significant if this membrane could manage to separate the homogeneous mixture of oil and water”, but it is a long way to get the target because it always is one of the top challenges in natural science up to now. In principle, if the pore size of our supramolecular frameworks becomes smaller enough to reach the molecular level, such a separation of homogeneous mixture will become possible, just like the metal organic frameworks (MOFs) using in gas separations.

Question 4: The author claimed the separation efficiency of both hydrophobic and hydrophilic liquids could reach over 99% for all measured mixtures. How was this datum of the separation efficiency experimentally obtained? Did the authors weigh the solvents after passing through the membrane, or by some method else? No detailed information on this part of the experiment was given in the paper.

Reply: Thanks for the comment. We added a description regarding separation efficiency

at the part of subtitle “Membrane devices, separation efficiency and flux calculation” on page 3 in the Electronic Supplementary Information (ESI). The separation efficiency (η) was determined by the equation: $\eta=(m/m_0)\times 100\%$, where m and m_0 are the weight of liquids to be separated before and after the separation. In the calculation, we ignored the part of adsorption because the membrane was wet before separation experiment. The separation for each sample was repeated five times and the average value was adopted, as presented in **Figure 4d**. The complemented data were added in Supplementary Figure 57.

Question 5: What is the effect of membrane fouled after each use?

Reply: Actually, as shown in Figure 4e, we evaluated the flux of the prepared membrane for the separation of water/ CCl_4 , ethylene glycol/ CCl_4 , and glycerin water solution (67.7%)/ CCl_4 after 10 cycles of switching (total 20 times of separation) because the flux change is in close correlation to the membrane stability and antifouling property. The data indicated that the flux for all samples showed almost no obvious decrease, revealing that the high stability of porous structure and anti-fouling characteristic since otherwise the flux value decreased or increased dramatically. Though the flux for glycerin water solution was pretty low in the present study due to its high viscosity, the membrane still maintained unfouled state due to the consistent flux. As we described in the part of mechanism discussion and mentioned above, the supramolecular framework in the fibrous assembly plays a key role in performing such a feature. The liquid to be separated can get into the nanopores comprising of framework structure to form a liquid layer on the surface of separation membrane, which protects the fouling from the other immiscible liquid to be

blocked. The formed liquid-liquid-solid three non-continuous phase decreases the membrane fouling efficiently. A corresponding discussion was added in page 7, last but line 4-5 of the first paragraph and the complemented evaluation based on the measurement of flux under different conditions such salt, acidic, and alkaline solutions was added in Supplementary Figure 58. An examination on sustainable stability for organic liquid and water was added in Supplementary Figure 53.

Question 6: With regards to the proposed mechanism, the author did nothing but static contact angle, which is absolutely not convincing.

Reply: Thanks for the comments. Since the membrane structure bears both hydrophobic and hydrophilic components and the changing over of oil and water is performed within the membrane device, it was hard to carry out an in-situ characterization via a simple experiment. Actually, beside static contact angle, we also used XRD (Figure 3c and 3d) to demonstrate the influence of hydrophobic and hydrophilic liquid wetting on the structure stretching and shrinking, which provided an indirect evidence to the proposed mechanism. In addition, a PDP simulation (Figure 5d and 5e, as well as corresponding data in ESI) was performed to support the analysis of the mechanism. In response to the question, we complemented some new data on contact angle and the surface energy estimation, as can be seen in Supplementary Table 3–4. The energy analysis of the result supports the mechanism understanding and the description is added in page 9 line 13-18, which is marked in yellow background. A subtitle “Surface model based on the minimization of system’s free energy” respect to the free energy and wetting pressure change is added in ESI.

For the proposed mechanism, the separation process can be simplified into two parts, the oil-water separation, and the joystick-modulated reversible switching. With the support of supplementary data, the mechanism discussion on the separation of oil and water can be further understood. Considering the similarity of present membrane structure type to that reported in publications (such as Aizenberg et al., Nature, 2011, 477, 443–447), the liquids to be separated, in our work, can be contained in the porous supramolecular framework and stabilized by capillary force because of the nano-sized pores (ca. 1.9 nm), hydrophobic or hydrophilic liquid can be stabilized. According to the decrease of free energy change (ΔE) for wettable liquids and the increase of wetting pressure (ΔP) (shown in Supplementary Table 3), the membrane is hydrophobic in the air state and can be wetted preferentially by hydrophobic liquids such as CCl_4 to form a stable liquid-infused layer. In contrary, the liquid layer could not be displaced by water. Hence, the water will be repelled within the given wetting pressure (ΔP).

From the added data (shown in Supplementary Table 4), we can also see that under the liquids, the calculated energy values (ΔE) of the membranes wetted by miscible liquids are always smaller than the one soaked with immiscible liquids. As a result, the immiscible liquids are blocked. Importantly, the calculated (ΔE) values under the liquids are always lower than those in air state while the pressure changes (ΔP) are higher than those in air state. Such surface property change further supports that the formation of liquid covering layer on the membrane surface and its enhanced effect to repel immiscible liquids. In the opposite state, the surface free energy analysis for the hydrophilic layer on membrane allows water to go through while the hydrophobic liquids will be blocked.

Base on the same principle, for the switchable separation process, the framework structure infused with joystick liquids can be easily transformed to hydrophobic or hydrophilic state, because the intermediate solvents are miscible with both liquid A and liquid B (shown in Supplementary Figure 55). Thus, the surface property of the membrane will be dominated by the liquids using in further wetting process, regardless of the polarity. Because the known model cannot be used for miscible liquid system, we carry out the dissipative particle dynamic (DPD) simulation to understand the switching process, as described in the part of main text regarding mechanism discussion.

Question 7: This paper is very badly written, and too many spelling mistakes and syntax errors appeared, for example (the errors listed as follows are only a small part, and there are more writing mistakes throughout the paper):

(1) ...exhibit vigorous materials potential applications...(“vigorous” in Line 1 in “Abstract”);

(2) ...processable conveniently because of their normal state...(“because” in Line 3 in “Abstract”);

(3) ...could be a preferable approach for structural flexibility...(“preferable” in Line 5 in “Abstract”);

(4) Ordered nanoporous structure ...exhibit...(Line 1 in “Abstract”, the subject word “structure” should be plural form.);

(5) ...during separation of immiscible liquids...(Line 9 in “Abstract”, it should be “...during the separation of immiscible liquids...”). Too many such kinds of mistakes appeared in the whole paper.

(6) ...structural symmetry, play a key role for diverse fantastic frameworks...(Line 3 in paragraph 1, “play a key role for...” should be “play a key role in....”)

(7) ...materials with suitable solvent trapping ability is an alternative manner...(Line 1 in paragraph 2, should be “are” instead of “is” because the subject word is the plural form)

Reply: Thanks very much for the kind corrections. We made a thorough polishing to the manuscript on English spelling, grammar, and expressions. Those improved places were marked in yellow background for reference.

Reply for Reviewer 2

General Comments: In this manuscript, the authors presented flexible supramolecular frameworks which are composed of inorganic polyanionic cluster, cationic organic host, and bridging guest. In the frameworks, the inorganic polyanionic cluster is connected with four organic cationic pillar[5]arene molecules by electrostatic interaction. And the organic cationic molecules are further used as a host to inclusion of the formed host complex with bridging nonionic guest. The resulting framework structures are able to form layered nanofibrous assemblies with enough structural rigidity and uniform porosity. Moreover, the framework structures could form supramolecular gels, which are favorable for processability. Interestingly, the surface properties of the porous frameworks can be modulated by filling hydrophilic/hydrophobic liquids, which renders the frameworks can be employed as a switchable oil-water separation membrane. The works are interesting and

important, and thus I recommend the acceptance of the manuscript after addressing the following minor issues.

Reply: Thanks very much for understanding the importance of this work.

Question 1: The authors should check all the Figure numbers. For example, the “Ionic complex (TBP)₄SiW” section at page 6 in the Electronic Supplementary Information, the “Supplementary Figure 16” should be changed to “Supplementary Figure 17”, and the original “Supplementary Figure 17” should be changed to “Supplementary Figure 18”.

Reply: We appreciate the important correction and we changed the citations and order of the Figures. Similar mistakes have also been corrected.

Question 2: In the application section, the methanol was used as joystick liquids. I want to know whether other solvents can be used as joystick liquids, such as DMF, THF, and DMSO.

Reply: Thanks for the important comment. Actually, ethanol, isopropanol, and THF can also be used for the same purpose, yet methanol seemed the best one from all common solvents because it is the only one that mixes all organic solvents and water. As summarized in **Supplementary Table 1**, DMF and DMSO can be used for switching the oil-water separations in principle, but we did not ascribe them to the joystick liquids due to their damage to the membrane. The reason is that they can dissolve the membrane by weakening the host-guest interaction and the interaction between framework layers.

Reply for Reviewer 3

General Comments: The manuscript describes the remarkable formation of a 2D, porous, supramolecular self-assembly made of hydrophilic, fully-oxidized Keggin-type POMs bearing the 4- charge and hydrophobic organic ligands for control of bonding and bridging between the assembled units. Contrary to the frameworks generated by coordinative or covalent bonds, the framework structure produced by the authors of this work is benefited from intermolecular interactions. The compounds are fully characterized and the results reported are adequately supported by the acquired data from different well-executed measurements. The applicability of organic-inorganic, POM-based materials in gel membranes for the controlled switching of oil-water separation has been demonstrated. In my opinion, this complete work deserves to be published in Nature Commun. after minor revision.

The authors should comment in the main text on:

Reply: Thanks for the positive comments and we made corrections accordingly.

Question 1: if the described self-assembly strategy can be extended to any other POM family, like e.g. Lindqvist, Anderson or Wells-Dawson structures?

Reply: Thanks for the very interesting question. Actually, we have carried out similar experiments for those clusters, and the results indicated that the formed supramolecular framework assemblies were in close relevant to the negative charges other than the type of inorganic clusters. For Lindqvist cluster, no framework structure formed since the two charges could not afford a crosslinking and instead of that linear structure became the main product. For Anderson type cluster with three negative charges, a hexagonal

framework structure could be predicted, yet due to the enlarged pore size in comparing to the square structure, the yielded higher flexibility made the obtained assembly unsuitable for a separation membrane unless the organic part is redesigned to match the change. As for the Dawson type clusters, their six charges make them possible for connecting organic cations to form cubic framework assembly. The relevant work is in progress.

Question 2: what is the highest POM-charge that still remains feasible for the construction of such supramolecular structures?

Reply: This is also an interesting question. Normally, for a three-dimension framework, the largest coordination number is six and thus, the anionic number of POMs should be six because more coordination numbers will lead to disorder. But in some cases, for the inorganic clusters bearing charges more than 6, it is also possible to obtain a cubic framework structure only if the ionically combined organic counterions maintain at 6, yet this will result in an incomplete charge replacement.

Question 3: what role does the Keggin-POM isomerism play for the framework assembly and its stability?

Reply: Thanks for the important question. In the present study, only size and charge of Keggin POM are demonstrated to influence the prepared supramolecular framework assemblies. Though there are several isomers for Keggin POMs, their charges maintain the same and the isomerism-induced small change in shape does not affect the ionic interaction and the orientation of organic cations under the driven of interfacial energy.

Question 1 on citation: Polyoxometalates as components of supramolecular assemblies.

Chem. Sci. 2019, 10, 4364-4376.

Reply: Thanks for the suggestion, we cited the literature as ref. 28

Question 2 on citation: Interplaying the amphipathic polyoxometalate interactions in solution and at solid–liquid interfaces: A toolbox for the technical application. *Nanoscale* 2019, 11, 4267–4277.

Reply: Thanks for the suggestion, we cited the literature as ref. 29

Question 1 on Typos: p.2: MOLDI-TOF  MALDI-TOF

Reply: Thanks for the helpful reminding. We have corrected it.

Question 2 on Typos: p.3: A reference doesn't appear for the sentence ending with "... the insertion of BCB into the cavity of TBP."

Reply: Thanks for the helpful reminding. We have added it as ref. 36.

Reviewers' comments:

Reviewer #1 (Remarks to the Author):

The revised manuscript addressed some of my previous concerns and the authors polished the language to make it more readable. However, it has been as long as two whole months since the first round of review, it seems that the author did little substantial improvement even after the revision. So I still strongly believe the quality of this manuscript (the novelty and the scientific significance) is far below typical Nature Communications articles and if it gets published it will affect the journal's reputation. My major concerns regarding this article are listed below.

1. In the "reply to question 1", I have questioned that why some organic solvents could dissolve the membrane while some others could not, and whether this phenomenon was due to some functional groups of organic solvent that led to this result. Unfortunately, the explanation the author had given was so confused and perfunctory that I can hardly find valid answers to my questions. Although the author gave a nearly one and a half page statement which was mainly focused on the three kinds of interactions: ionic interaction, host-guest inclusion, and interlayer interaction. However, what was the final conclusion then? If the author had confidence in his theory, necessary control experiments should be, at least, included to prove it, for example trying some more different solvents to verify the theory.

2. In the "reply to question 2", the author "do not with the comment" and repeatedly claimed the novelty of this work. Actually, from a material point of view, superhydrophilic / superoleophobic hydrogel-coated mesh membranes that achieved excellent oil/water separation under gravity have already been reported many years ago, for example Jiang's work in 2011 (10.1002/adma.201102616). Besides, some other similar works can also be found, such as Chemical Engineering Journal 338 (2018) 271-277; 10.1038/srep02326; Chem. Sci., 2013, 4, 591; Nanoscale, 2016, 8, 7638, and so on. And most of these reported works can not only realize the separation of immiscible liquid systems but even can achieve the separation of homogeneous oil/water system, for example oil/water emulsion, which would be absolutely more significant in practical applications.

3. With regard to the question 3, my concerns, in short, included actually two aspects:

(1) Only the separation of immiscible liquids is not that significant because the two species of the liquids are intrinsically immiscible and consequently located in a different layer.

This concern, however, was immediately denied by the author in his reply: "The comment seemed correct but it was not true...". Indeed, I agree with what the author argued—"In many actual systems, oil and water do not exist in only two layers with a clear interface. Instead of that, the bulk oil and water mixtures are often cut into a great many small biphasic domains which cannot be separated as easily as ideal two phases that the comment considered". However, none of any control experiments have been added to prove its potential separation ability in such an actual system. After all, to separate oil and water which does not exist in only two layers with a clear interface is more difficult than to separate immiscible layered liquid system. It seems that the author knows everything but does nothing practically, except for empty arguments to support his theory and to reply a reviewer.

(2) It will be more significant if this membrane could manage to separate the homogeneous mixture of oil and water.

As for this suggestion, the author explained that "it is a long way to get the target because it always is one of the top challenges in natural science up to now". Here, I'm curious that is it really that difficult to do some homogeneous mixture of oil and water? It seems that the author apparently has a weak knowledge of the field of oil/water and literature research is quite insufficient. Because there have been numerous works related to the separation of homogeneous oil/water system, for example:

<https://doi.org/10.1016/j.seppur.2018.10.001>.

J. Mater. Chem. A, 2014, 2,2445

ACS Appl. Mater. Interfaces 2014, 6, 16204-16209

Adv. Mater. 21, 3601-3604 (2009).

NPG Asia Materials (2014) 6, e101; doi:10.1038/am.2014.23

J. Mater. Chem. A, 2013, 1,14071
RSC Adv., 2017, 7, 9051
ACS Appl. Mater. Interfaces 2018, 10, 30860-30870
...and so on.

And what is the most strange is that the author claimed that in principle, if the pore size of our supramolecular frameworks becomes smaller enough to reach the molecular level. Now that the author realized that this membrane cannot achieve the separation of this level, why did he at the same time argue that "In many actual systems, oil and water do not exist in only two layers with a clear interface." ?

Reviewer #2 (Remarks to the Author):

The issues have been addressed, and thus the manuscript can be accepted.

Reviewer #3 (Remarks to the Author):

The authors have adequately addressed my questions and concerns. In my opinion, this manuscript can now be considered for publication in Nature Commun. in its present form.

Responses to the Reviewers' Comments

For Reviewer 1

Comment: The revised manuscript addressed some of my previous concerns and the authors polished the language to make it more readable. However, it has been as long as two whole months since the first round of review, it seems that the author did little substantial improvement even after the revision. So I still strongly believe the quality of this manuscript (the novelty and the scientific significance) is far below typical Nature Communications articles and if it gets published it will affect the journal's reputation. My major concerns regarding this article are listed below.

Reply: In the first revised version of the manuscript we carried out a lot of revisions in the main text including the suitable modification to the motivation, novelty and the advantages. Of course, we also addressed the general situation of oil/water separation like emulsion system, as was explained in the replies to the previous comments. The unique property of the supramolecular fibers as a complete synthetic system was that the new kind of materials can be used for the separation membrane and all of the components as well as the formed framework pores were demonstrated to play important roles in both the separation and in-situ switching of oil and water. We also complemented several data in the supplementary information (SI) to demonstrate those explanations. Apparently, the full artificial designed assemblies provide wider opportunities for the separation membrane possessing extendable functions such as microemulsion separation, nano-particle separation, and protein separation in the ongoing work. Some of them were indeed

observed applicable and the work is in progress.

Question 1: In the “reply to question 1”, I have questioned that why some organic solvents could dissolve the membrane while some others could not, and whether this phenomenon was due to some functional groups of organic solvent that led to this result. Unfortunately, the explanation the author had given was so confused and perfunctory that I can hardly find valid answers to my questions. Although the author gave a nearly one and a half page statement which was mainly focused on the three kinds of interactions: ionic interaction, host-guest inclusion, and interlayer interaction. However, what was the final conclusion then? If the author had confidence in his theory, necessary control experiments should be, at least, included to prove it, for example trying some more different solvents to verify the theory.

Reply: This is an important question regarding the structural stability of whole framework assemblies rather than a simple influence of function group of solvent molecules. Since the framework assembly in the formed supramolecular fibers involved multiple intermolecular combinations from electrostatic, interlayer, and host-guest interactions, we outlined the main reasons whether the used solvents/liquids could destroy the separation membrane in the previous response. Actually, more than 30 organic solvents and liquids were employed to pass through the separation membrane and to evaluate their influence on membrane structure, as summarized in the previous Supplementary Table 1 and Table 2 in SI. The electrostatic interaction was replied to maintain stable under all conditions in the present study. The newly complemented Supplementary Figure 60 in SI further confirmed the stability of the ionic interaction in a strong polar solvent like DMSO. The

solvent influence on the latter two types of interactions is complicated and we re-evaluated the solvent dissociation to the interaction between framework layers. Fortunately, from the facts that the membrane dissolved in DMSO but the host-guest interaction was not affected obviously in it (Supplementary Figure 59), a new understanding is that the interlayer interaction can be broken in such a strong polar solvent. In the present stage, however, it is still difficult to give a direct evidence to demonstrate what triggered the decomposition of interlayer interaction. A preliminary explanation is that the solvent polarity and the solubility of isolated (TBP)₄SiW and BCB units should be the important factors.

From Supplementary Table 1 in SI, both strong polar and weak polar organic solvents were found to dissolve the membrane or make the membrane shrink. The inclusion force of pillararene group with guest was demonstrated to source mainly from the dipolar-dipolar, C-H... π , and hydrophobic interactions (Chem. Commun., 48, 2967–2969 (2012); Angew. Chem. Int. Ed., 50, 1397–1401 (2011); J. Am. Chem. Soc., 137, 1440–1443 (2015)). Thus, the solvent polarity, molecular size, and function group are related to the membrane stability by weakening or breaking the host-guest interaction of (TBP)₄SiW@BCB. In the case of the weak polar solvents like CH₂Cl₂, CH₂Br₂, Br(CH₂)₃Br, Br(CH₂)₄Br, and NC(CH₂)₄CN, their dissolution for the membrane can be ascribed to the competitive inclusion-induced dissociation of initial guest BCB, which broke the crosslinked framework structure. Beside the supports from publications (J. Phys. Chem. B 2015, 119, 6711–6720; Chem. Commun. 2014, 50, 12420–12433) regarding solvent polarity and the influence of function groups, the observed changes of host-guest

interaction of $(\text{TBP})_4\text{SiW}@\text{BCB}$ in deuterated DMF, and CD_2Cl_2 , shown in previously complemented Supplementary Figure 59 in SI provides experiment evidence. The appearance of proton peaks ascribing to bridging guest BCB at non-included state demonstrated the partially breaking up of original host-guest inclusion. In the presence of adiponitrile, due to the enhanced interaction of cyano groups with $(\text{TBP})_4\text{SiW}$ host, strong replacement of bridging guest BCB was observed. In short, the function groups of solvents are possible to play an important role for dissolving the membrane by either weakening the inclusion interaction between $(\text{TBP})_4\text{SiW}$ and BCB or the interlayer interaction but the detailed mechanism for how the groups worked has to be verified in the further investigations. The corresponding paragraph on this question was rewritten accordingly, as shown in page 6 line 24–41 marked in yellow background.

Question 2: In the “reply to question 2”, the author “do not with the comment” and repeatedly claimed the novelty of this work. Actually, from a material point of view, superhydrophilic / superoleophobic hydrogel-coated mesh membranes that achieved excellent oil/water separation under gravity have already been reported many years ago, for example Jiang’s work in 2011 (10.1002/adma.201102616). Besides, some other similar works can also be found, such as Chemical Engineering Journal 338 (2018) 271-277; 10.1038/srep02326; Chem. Sci., 2013, 4, 591; Nanoscale, 2016, 8, 7638, and so on. And most of these reported works can not only realize the separation of immiscible liquid systems but even can achieve the separation of homogeneous oil/water system, for example oil/water emulsion, which would be absolutely more significant in practical applications.

Reply: All the mentioned results showed efficient separation for oil/water, yet only Jiang's group (*Adv. Mater.* 2011) and Zhang's group (*Chem. Eng. J.* 2018) reported the polymer hydrogel-coated steel mesh for separation. Other publications dealt with the coatings of zeolite (*Chem. Sci.* 2013), silicate/TiO₂ (*Sci. Rep.* 2013), and carbon black/SiO₂ (*Nanoscale* 2016) on a stainlessness steel mesh via different deposition procedures. In our work, instead of coating on the screen wire of steel mesh, the prepared supramolecular gel fibers comprising of small molecular components were spread on the support acting as the independent membranes. Therefore, the prepared materials themselves used here are different from those in publications and the formed supramolecular gel fibers can form an independent layer of separation membrane. Up to date, there are no reported results dealing with the separation membrane composed of small molecule organic gels as far as we know. In addition, with the new framework structure design, the in-situ reversible switching process can be realized conveniently. Therefore, the reported results displayed a contribution for supramolecular polymer in two-dimension framework structure. As explained in the previous response to the question, the extension of the current separations from present bulk oil/water system to emulsion and microemulsion systems can be expected. Furthermore, the separation of inorganic nanoparticles and even proteins will become applicable after further improvement of the membrane structure by mixing with additional polymers.

Question 3: With regard to the question 3, my concerns, in short, included actually two aspects:

(1) Only the separation of immiscible liquids is not that significant because the two species

of

the liquids are intrinsically immiscible and consequently located in a different layer.

This concern, however, was immediately denied by the author in his reply: "The comment seemed correct but it was not true....". Indeed, I agree with what the author argued—"In many actual systems, oil and water do not exist in only two layers with a clear interface. Instead of that, the bulk oil and water mixtures are often cut into a great many small biphasic domains which cannot be separated as easily as ideal two phases that the comment considered". However, none of any control experiments have been added to prove its potential separation ability in such an actual system. After all, to separate oil and water which does not exist in only two layers with a clear interface is more difficult than to separate immiscible layered liquid system. It seems that the author knows everything but does nothing practically, except for empty arguments to support his theory and to reply a reviewer.

(2) It will be more significant if this membrane could manage to separate the homogeneous mixture of oil and water.

As for this suggestion, the author explained that "it is a long way to get the target because it

always is one of the top challenges in natural science up to now". Here, I'm curious that is it really that difficult to do some homogeneous mixture of oil and water? It seems that the author apparently has a weak knowledge of the field of oil/water and literature research is quite insufficient. Because there have been numerous works related to the separation of

homogeneous oil/water system, for example:

<https://doi.org/10.1016/j.seppur.2018.10.001>.

J. Mater. Chem. A, 2014, 2,2445

ACS Appl. Mater. Interfaces 2014, 6, 16204-16209

Adv. Mater. 21, 3601-3604 (2009).

NPG Asia Materials (2014) 6, e101; doi:10.1038/am.2014.23

J. Mater. Chem. A, 2013, 1,14071

RSC Adv., 2017, 7, 9051

ACS Appl. Mater. Interfaces 2018, 10, 30860-30870

...and so on.

And what is the most strange is that the author claimed that in principle, if the pore size of our supramolecular frameworks becomes smaller enough to reach the molecular level.

Now that the author realized that this membrane cannot achieve the separation of this level, why did he at the same time argue that "In many actual systems, oil and water do not exist in only two layers with a clear interface." ?

Reply: For (1), both the separations of simple immiscible biphasic system and emulsion/microemulsion system are believed important in lab use and industry, because the former can be used as a simplification of complicated separation systems and a thorough separation of simple immiscible liquids is also required though the latter represents a more regular case. We have explained in the previous response that the present study as a strong base can be extended to the separation of emulsion/microemulsion by mixing the current membrane with a commercial polymer due

to the interface compatibility. The utmost scale limit for a microemulsion droplet can be reached to about 2 nm in size. The obtained preliminary results strongly supported the supramolecular framework fibers for the membranes constructed by the supramolecular framework fibers showing wide potentials in various separations.

For (2), it seemed that the previous reply lead to a misunderstanding and the reason may come from the comprehension for the phrase “homogeneous oil/water system” in the previous comments. The “homogeneous mixture” for us means the true solution like the mixture of methanol and water but in the present question it seemed to be an emulsion or a microemulsion, which we believed they are two-phase dispersion systems as well. With this understanding difference, we gave the previous explanation for the question because the true mixture solutions without phase interface are really difficult to be separated via a simple filtration approach without a precise design for the porosity of membranes. As for the emulsion/microemulsion systems, there are of course many publications through diverse techniques and several important published results were cited in the first submission (ref. 25, 27, 45, 23). Following the current study, consecutive switching separation of different microemulsions with thermodynamic stability will be achieved based on the current supramolecular fiber membranes.

For Reviewer 2

No response is required.

For Reviewer 3

No response is required.

REVIEWERS' COMMENTS:

Reviewer #1 (Remarks to the Author):

I just read ref 24 by prof. Jiang et al. "Xue, Z. X. et al. A novel superhydrophilic and underwater superoleophobic hydrogel-coated mesh for oil/water separation. Adv. Mater. 23, 4270–4273 (2011)". The separation efficiency mentioned in current paper has been achieved almost ten years ago, based on a similar system. Indeed I just randomly choose this one, and there are already lots of similar reports. I do not believe the novelty of current research reaches the high criterion expected for a journal like Nature Communications.

Reviewer #3 (Remarks to the Author):

I agree with Reviewer 1 who says in the part "Question 3 / aspect 1":

"However, none of any control experiments have been added to prove its potential separation ability in such an actual system. After all, to separate oil and water which does not exist in only two layers with a clear interface is more difficult than to separate immiscible layered liquid system. It seems that the author knows everything but does nothing practically, except for empty arguments to support his theory and to reply a reviewer."

The authors Reply as follows: "... We have explained in the previous response that the present study as a strong base can be extended to the separation of emulsion/microemulsion by mixing the current membrane with a commercial polymer due to the interface compatibility. The utmost scale limit for a microemulsion droplet can be reached to about 2 nm in size. The obtained preliminary results strongly supported the supramolecular framework fibers for the membranes constructed by the supramolecular framework fibers showing wide potentials in various separations."

Thus, the results obtained are only preliminary and the authors need to provide convinced control experiments to prove the potential separation ability in the studied system. After this Major revision, the study might be considered for publication in Nature Commun.

Reply to the Reviewers' Comments

For Reviewer #1

Comments: I just read ref 24 by prof. Jiang et al. "Xue, Z. X. et al. A novel superhydrophilic and underwater superoleophobic hydrogel-coated mesh for oil/water separation. *Adv. Mater.* 23, 4270–4273 (2011)". The separation efficiency mentioned in current paper has been achieved almost ten years ago, based on a similar system. Indeed I just randomly choose this one, and there are already lots of similar reports. I do not believe the novelty of current research reaches the high criterion expected for a journal like *Nature Communications*.

Reply: As we mentioned in the last reply, the gel-coated membranes have been indeed reported for oil/water separation, but the switchable process is still not realized in this type of system. We developed a new kind of supramolecular organic gel (not polymer hydrogel) fibers with flexible two-dimensional frameworks that represent a fully designed porous material. While strengthening the gel fibers greatly, with the unique nanopores in the formed supramolecular framework, the gel membrane displayed in situ switchable liquid separation. The consecutive transformation through a liquid infusion approach for controllable oil/water separation is still desired in many aspects. We also mentioned the direct extendibility of the prepared framework fibrous membrane for emulsion separations in the previous replies, based on the synthetic pores.

For Reviewer #3

Comments: I agree with Reviewer 1 who says in the part "Question 3 / aspect 1": "However, none of any control experiments have been added to prove its potential separation ability in such an actual system. After all, to separate oil and water which does not exist in only two layers with a clear interface is more difficult than to separate immiscible layered liquid system. It seems that the author knows everything but does nothing practically, except for empty arguments to support his theory and to reply a reviewer."

The authors Reply as follows: "... We have explained in the previous response that the present study as a strong base can be extended to the separation of emulsion/microemulsion by mixing the current membrane with a commercial polymer due to the interface compatibility. The utmost scale limit for a microemulsion droplet can be reached to about 2 nm in size. The obtained preliminary results strongly supported the

supramolecular framework fibers for the membranes constructed by the supramolecular framework fibers showing wide potentials in various separations." Thus, the results obtained are only preliminary and the authors need to provide convinced control experiments to prove the potential separation ability in the studied system. After this Major revision, the study might be considered for publication in Nature Commun.

Reply: We do work continuously on the further extensive potentials concerning the separation membrane for emulsion and nanoscale entities and the experiment results supported the argument. Considering the possible deviation to the consistency and the main topic on the fabrication of flexible two-dimensional supramolecular frameworks in organic gel fibers, we did not add these experiment results as the control separation evidence in the previous submission. Here, to demonstrate that the synthesized materials and the used method are applicable and extendable in a simple way, we provided some cogent data shown in the following, instead of main text, for your reference.

[redacted]